# SurDis: A Surface Discontinuity Dataset for Wearable Technology to Assist Blind Navigation in Urban Environments

**Kuan Yew Leong**
A.I. System Research Co., Ltd.
Kyoto, 606-8302 Japan
`kuanyew.leong@gmail.com`

**Siew Mooi Lim**
Tunku Abdul Rahman
University of Management and Technology
`siewmooi@tarc.edu.my`

## Abstract

According to World Health Organization, there is an estimated 2.2 billion people with a near or distance vision impairment worldwide. Difficulty in self-navigation is one of the greatest challenges to independence for the blind and low vision (BLV) people. Through consultations with several BLV service providers, we realized that negotiating surface discontinuities is one of the very prominent challenges when navigating an outdoor environment within the urban. Surface discontinuities are commonly formed by rises and drop-offs along a pathway. They could be a threat to balancing during a walk and perceiving such a threat is highly challenging to the BLVs. In this paper, we introduce SurDis, a novel dataset of depth maps and stereo images that exemplifies the issue of surface discontinuity in the urban areas of Klang Valley, Malaysia. We seek to address the limitation of existing datasets of such nature in these areas. Current mobility tools for the BLVs predominantly focus on furniture, indoor built environments, traffic signs, vehicles, humans and various types of objects' detection above the surface of a pathway. We emphasize a specific purpose for SurDis – to support the development of assistive wearable technology for the BLVs to negotiate surface discontinuity. We consulted BLV volunteers on the specifications of surface condition that could become hazardous for navigation using 3D printed replicas of actual scaled-down scenes, and identified locations that are frequented by the BLVs as our target data collection fields. With feedback from these volunteers, we developed a lightweight, small and unobtrusive prototype equipped with a tiny stereo camera and an embedded system on a single board computer to capture the samples from 10 different locations. We describe instrument development, data collection, preprocessing, annotation, and experiments conducted. The dataset contains: (1) more than 17000 depth maps generated from 200 sets of stereo image sequences, (2) annotations of surface discontinuity in the depth maps, and (3) bitmap stereo image pairs corresponding to the depth maps in (1).

## 1 Introduction

Safe and efficient navigation has profound importance to the BLVs, as it might be the prerequisite for getting one to achieve many social functions. Through consultations with several local BLV service providers in Malaysia, it is realized that there is a diverse range of navigational challenges faced by the BLVs. To achieve safe and effective navigation, the BLVs need to access global information relevant to orientation and position on one hand, and deal with local threats along their pathway on the other. Negotiating surface discontinuities is one of the very prominent challenges when navigating an outdoor environment within urban areas. Surface discontinuities are various kinds of rises and

36th Conference on Neural Information Processing Systems (NeurIPS 2022) Track on Datasets and Benchmarks.

drop-offs along a pathway (Geruschat & Smith, 2010). The rises and drop-offs change the gradient of the navigational surface. They could be a threat to balancing during a walk, and perceiving such a threat is a difficult challenge to the BLVs without some proper aids (Kuyk et al., 2004; Goodrich & Ludt, 2002).

If the presence of universal access facilities such as tactile ground surface indicators, handrails along staircases, pedestrian ramps, subways, properly covered drainages and et cetera is a sign of equipping the BLVs with better built environment, it is a little-known reality that even in some modern cities, most of such facilities are often only available around limited public transportation infrastructures, certain well-planned urban landscapes and some government or private properties (Hussein & Mohd. Yaacob, 2013). Additionally, traditional navigation aids such as a guide cane might not always be helpful when the built environment does not comply with BLV accessibility design standards. Uneven staircases, steep drop-offs, uncovered drainage, absence of terminal posts between junctions, stairways without proper handrails and other types of surface discontinuity are common hazards to the BLVs due to poor enforcement or implementation of regulations for accessible design. This issue is especially prominent in low to middle income countries, and it is found to be a common scenario around some busy urban areas in the Klang Valley of Malaysia. Some of these places offer major private businesses and public services, and hence safe accessibility to them is important to the BLVs.

Surface discontinuities such as staircases, small steps, joints between walkways and curbs are needed for the continuity of navigation. However, some of them might be hazardous due to unregulated circumstances. Figure 1 shows an illustration of such surface discontinuities, and some samples we found at several urban areas in Malaysia. Hazardous conditions are indicated by the red arrows, on the right image (counterclockwise) we have: partially covered drainage next to some steps that connect the walkway, uncovered drainage between a steps connecting the road and the aisle, high altitude drop-off leading to uncovered drainage at the edge of walkway, uncovered drainage between a ramp and some steps, drop-off along the edge of a walkway without railing, and blended gradients between a ramp and steps.

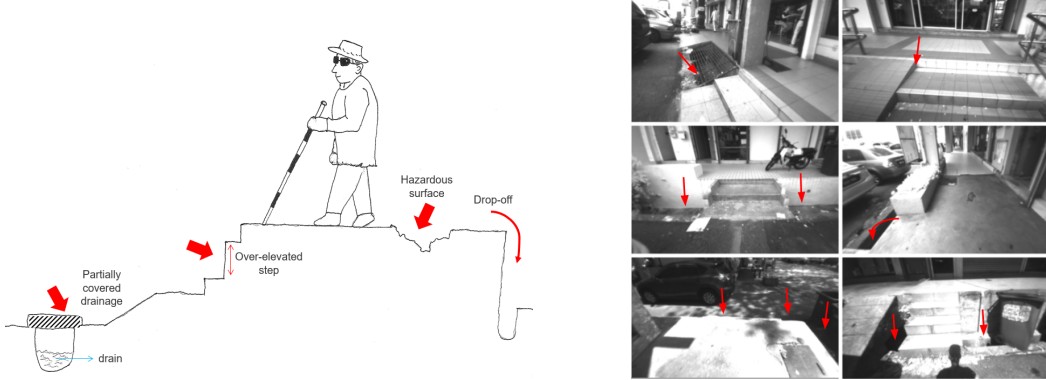

Figure 1: Left: An illustration of surface discontinuities faced by the BLVs. Right: Examples of hazardous surface discontinuity found at several urban areas within Malaysia.

Most of the published works that can be adopted for blind navigation have predominantly focused on solving the problem of obstacles detection above the surface of a pathway. For examples, they detect either static objects such as furniture and other indoor objects (Takizawa et al., 2013), stairs (Vlaminck et al., 2013; Pérez-Yus et al., 2015; Dang et al., 2016; Westfechtel et al., 2018) and traffic signs (Yuanlong & Mincheol, 2014), or dynamic objects such as bicycles (Young et al., 2016) and pedestrians (Sun et al., 2018; Jeon et al., 2019 ). However, there is very limited research done for blind navigation in negotiating surface discontinuity at outdoor urban environments, let alone publicly available dataset that tailored for such research. Some of these datasets mentioned in their works might not be suitable for developing wearable assistive tools for the BLVs because the images were captured very differently from how the BLVs would be positioning themselves on a pathway. There were some research works conducted to address the global problems of geo-positioning and wayfinding for the BLVs (Fernandes et al., 2012; Karimi, 2015; Serrão et al., 2015), but in contrary, little attention was given to the local problem like negotiating surface discontinuity of a pathway.

Apart from the lack of research works addressing surface discontinuity, datasets that could exemplify such issue are equally limited. There are several publicly available datasets focusing on the urban pathway such as the Málaga urban dataset (Blanco et al., 2014), CrowdDriven (Jafarzadeh et al., 2021), and UrbanNav (Weisong et al., 2020). However, these datasets were mainly captured by image sensors on vehicles and they are meant for autonomous vehicles on the road. (Wigness et al., 2019) published Robot Unstructured Ground Driving (RUGD) dataset with video sequences of unstructured environments. The dataset exemplifies pathways in some natural settings and it was intended for autonomous vehicles. This dataset can potentially be adopted for blind navigation in natural environments, but it is not relevant to urban built structures. Cityscapes Dataset (Cordts et al., 2016) focuses on semantic understanding of urban street scenes of 50 cities with 30 classes of objects i.e. road, sidewalk, parking, bridge, person and terrain. This is close to the urban scenario for blind navigation that we are proposing, but apart from the classes "sidewalk, ground, and rail track" there is no classes that distinguish between surface discontinuity and smooth pathway.

The Project Sidewalk (Eisenberg et al., 2022) is an on-going project started in the US since 2012 to crowdsource sidewalk images using online map imagery. They have a broader purpose to collect street-level accessibility data from every street in the world and enable development of location-based technologies for accessibility. Their method of acquiring samples are based on volunteers to annotate and label images from online maps. While such dataset might be large and general enough for the development of various accessibility technologies, there is no guarantee that these images are specific to the issue of surface discontinuity in blind navigation. Images from Project Sidewalk might not be in proper perspective from the forward-facing point of an BLV person during navigation.

VIsual Dataset for Visually Impaired Persons (VIDVIP) (Tetsuaki, 2021) offers the most relevant images for blind navigation in urban environments. It has 538,747 instances for 32,036 images in 39 classes of labels including person, wall, bicycle, door, elevator, signboard, stairs, steps, handrail, crosswalk, traffic light, safety cone and several others. Again, VIDVIP does not offer any labels of surface condition. Furthermore, this dataset was meant to support the BLVs in Japan specifically, thus all samples were collected within Japan only. The pathways for pedestrian in Japan are typically well-built and adhered to strict regulations hence the issue of hazardous surface condition is rare. This makes it inappropriate for our local (Malaysia) usages as each country has its unique built environment and standards.

Taking into account the gaps and aforementioned motivation for assistive tool in negotiating surface condition for blind navigation, we present SurDis - the **Sur**face **Dis**continuity Dataset of urban areas in Malaysia. SurDis has 200 sets of depth map sequences with annotation of various surface discontinuities from 10 selected locations, captured in video recording mode by a person mimicking the walking style of a typical BLV individual. Each sequence set contains about 100 to 150 depth maps, and we generated a total of 17302 such depth maps. We also provide the original stereo image sequences corresponding to the depth maps. The justification for depth maps (or stereo images) over single images is that the former can offer better distant estimation of possible surface hazards. This extra information can become helpful in developing assistive tools for the BLVs. The documentation and download link of SurDis dataset can be found on this site: https://github.com/kuanyewleong/surdis.

## 2   Dataset Generation

The local BLV service providers were consulted at the early phases to identify the problem, understand more about the limitation of a guide cane in addressing surface discontinuity, and communicate the specifications of surface condition related to blind navigation. It is through these consultations that the following insights were gained and benefited our data generation.

### 2.1   Getting Specifications of Surface Discontinuity through 3D Printing

One of the most important items before data collection was to understand more precisely the types of surface discontinuity that is deemed hazardous by the BLVs during their outdoor navigation. It was impossible to visually show the BLVs some pictures or videos of surface discontinuity sampled from the field. Thus, an alternative approach was to let them touch and interpret some 3D printed replicas of the surface discontinuities, where they could form some mental pictures of the samples. During this pre-data collection stage, such communication helped the research to identify the right targets to be sampled from the field. Figure 2 shows an example of a potential sample, with its

image taken from the field, 3D model and the 3D printed replica. All the 3D replicas used in the survey were proportionately scaled down from the actual measurements taken from the field. We observed about 9 distinct types of pathway's surface around the targeted urban areas, which could be differentiated by a sighted person based on their physical attributes. These 9 types of surface were smooth walkway, down-steps, up-steps, drop-off, rise, down-ramp, up-ramp, drainage along pathway, and mixed-gradient (this is usually the meet-up point of several different terrains). We constructed 3 replicas each for these 9 types of surface, and thus we had a total of 27 replicas for the survey.

During the consultation, these 3D printed replicas were presented to a group of BLV volunteers and they interpreted the details they could touch and sense with their fingers. Based on the feedback, it was found that most of the surface discontinuities that could potentially be hazardous to the BLVs, and difficult to tackle by a guide cane are uncovered drainage along a pathway, sharp drop-off without safety handrail, unevenly built up- and down-steps (they had no issues with steps that were properly built), and mixed-gradient.

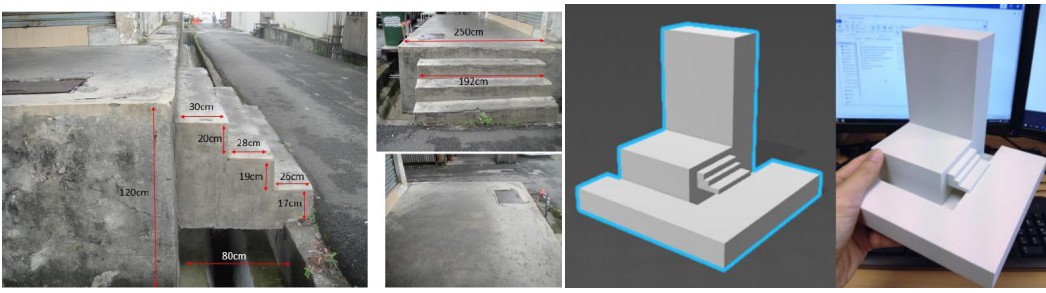

Figure 2: From left to right: Three different angles for a sample snapped from the field, its 3D model and printed replica. The 3D replica is proportionately scaled down from the ground truth.

## 2.2    Instrument Development and Setting

We developed a lightweight, small and unobtrusive wearable prototype as the data collection instrument with consultation from BLV volunteers. The BLVs would not abandon their guide canes while adopting some new technologies. The stereo camera must be set at a position that experiences the least effect from user's body movement. Based on Rodriguez et al. (2012) and Pérez-Yus et al. (2015) as well as feedback form our volunteers, the chest area of a person could be a strategic location for camera sensor. This area is less affected by the body movement when one is walking in a linear direction. To operate the sensor and other processing tasks in the form of a wearable prototype, a small computer platform was required. Our sensor is a factory calibrated stereo camera known as DUO MLX R2 developed by Code Laboratories. It has industrial grade monochrome global shutter sensors, and fully programmable active LED array. Other specifications of DUO MLX R2 can be found here: https://duo3d.com/docs/articles/duo-mlx. We used Odroid-XU3 (a single board computer developed by Hardkernel) for the processor. Specifications of Odroid-XU3 can be found here: https://www.hardkernel.com/shop/odroid-xu3/.

The camera was tuned to point at approximately 35 degrees facing downward from the horizon. At this angle, the camera could record proximal drop-offs without much issues. With 170 degrees of field of view from the camera, it can capture both proximal and distant scenes. We are most interested in the proximal scene as this is the common region we might capture the surface of a pathway. Under this setting, the user is approximately 80 centimeters away from the region of interest. Figure 3 shows the actual setup of the instrument. The camera was set to capture the highest allowable resolution of images at 752 x 480 pixels, and the raw images were uncompressed in bitmap format. An algorithm was implemented to take care of the auto-exposure of the camera as the data collection would be done at outdoor environment, where the amount of sunlight could be varied from time and places. Table 1 summaries our camera configurations for image sequence recording.

## 2.3    Data Sampling

Using the developed instrument, we collected over 200 sets of stereo image sequences from 10 different locations. The sampling method employed was judgmental sampling, which is a non-

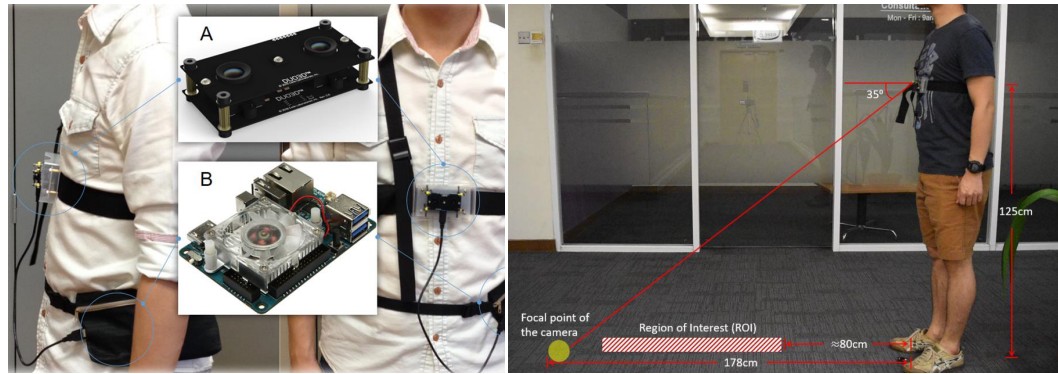

Figure 3: Left: The data collection instrument: the processor (B) was embedded into a waistpouch together with a lithium battery as the power source, while the camera (A) was set at chest level. Right: camera setting and region of interest

Table 1: Stereo camera configurations during data capturing.

| Specification | Value |
| --- | --- |
| Image file format | Bitmap (BMP) |
| Colour mode | Monochrome |
| Resolution | 752 x 480 pixels |
| Frame rate | 30 fps |
| Pixel format | 8-bit integer (0 – 255) |

probability method based on justification that certain areas could have more samples as compared to others. This is true in most cases because certain urban areas were newly developed and thus new set of building regulations were strictly followed, while certain areas with older buildings or walkways might not have followed proper regulations at the time when they were built. We also factored in the clues given by our BLV volunteers. Eventually, we collected samples from the following locations where high frequency of targeted surface discontinuity were found. These locations are Damansara Perdana, Kota Damansara, Bandar Utama, Up-Town, Damansara Jaya, Bandar Sunway – PJS 7, Bandar Sunway – PJS 9, Bandar Sunway – PJS 10, Brickfields and Pudu.

## 2.4 The Dataset

An orientation and mobility (O&M) specialist from Vision Australia (Dandenong) had demonstrated the formal usage of different types of guide cane for blind navigation to us during a consultation session. The O&M specialist had also personally conducted a short training for us on the guide cane usage in some urban outdoor environments. With this knowledge, we armed with a guide cane and walked in the style of a BLV person during capturing of the image sequences. The guide cane is mostly not present in the images as it is proximal to the body and out of the field of view of the camera. The image sequences were captured in short linear navigation by a sighted person along the walkways. Each sample is a pair of left and right images (hence stereo vision) sequence of a walkway connected to an incoming surface discontinuity. A sample could contain approximately 50 to 150 image pairs based on the duration of the recording at the field. They were collected during sunny days under direct sunlight or indirect sunlight depending on the locations. The recording tasks were performed under natural (uncontrolled) environment hence some samples might contain pedestrians or vehicles (which were anonymized). We used only one single person to put on the wearable sensor for all samples collection to ensure the consistency of SurDis.

### 2.4.1 Pre-processing and Anonymizing

To achieve quality dataset, we conducted cleaning and whitening on the collected image samples. Sorting out noise was the major cleaning task. With careful inspection of the raw data, it was found that the digital noise in the images was very little, except some fix-pattern noise. The occurrence

of this noise was random, and it typically had a bended grey and white banding covering some large areas of the images. The images with noise was removed upon manual inspection. Apart from digital noise, there were also large amounts of irrelevant instances within some images. These instances were image sequences captured at the beginning and/or ending of the recording. They were captured because at the beginning of recording, sometimes the camera was not pointing to the intended direction. This happened at the end of some recording too, we often turned to some random directions before the device halt its operation, thus captured some irrelevant scenes by accident. These irrelevant images were all removed. We then checked for pedestrians, vehicles and property names, and proceed to anonymize them using the blur tool in GIMP (GNU Image Manipulation Program).

Next, we conducted Principal Component Analysis (PCA) whitening to the images as a mean to normalize them. Firstly, this was needed because the images were captured under natural lighting and they vary a lot due to different sunlight amount on different days. Secondly, the auto-exposure of the camera wasn't always optimized causing some variation even within the same image sequence. A whitening process (or transformation) is a linear transformation that converts the feature vectors based on a covariance matrix into a new set of vectors whose are uncorrelated and have a variance of value 1 for each of them. As such, it is generally a beneficial practice to pre-process the data with whitening, since it de-correlates the data and makes them easier to model. However, we remained cautious that this point is arguable as Koivunen and Kostinski (1999) pointed out that its benefit is dependent on the data and its subsequent processes.

In the first step of PCA whitening, Singular Value Decomposition (SVD) was applied to compute the eigenvector decomposition from the zero-centred data. SVD uses a matrix analysis that decomposes a high dimensional matrix to a low dimensional representation. This makes it easy to eliminate the less important components of that representation and produce any desired number of dimensions from the elimination. The image resolution is 752 x 480 pixels, and this would eventually produce a dimension of 360960 in a single channel. It is crucial to preserve as much information as possible from the singular values. A calculation on total energy from the singular values was performed to retain the components that contribute to 90% of the information. We performed several trials and observed that 292380 out of the 360960 components preserve about 90% of the information. Based on the select 292380 singular values, the other low singular values were zeroed out. Now with the remaining eigenvectors, the rotated version of the data was computed by the product of inversed eigenvectors to the original pixel matrix. Finally, the PCA whitened data, was computed using Equation (1),

$$whitenedX = diag(1/\sqrt{(diag(singularvalues) + \epsilon)}) * xRotate \qquad (1)$$

in which epsilon is a small constant to prevent division by zero. Figure 4 shows three patches of image before the whitening (top), and the result (bottom) after their whitening process.

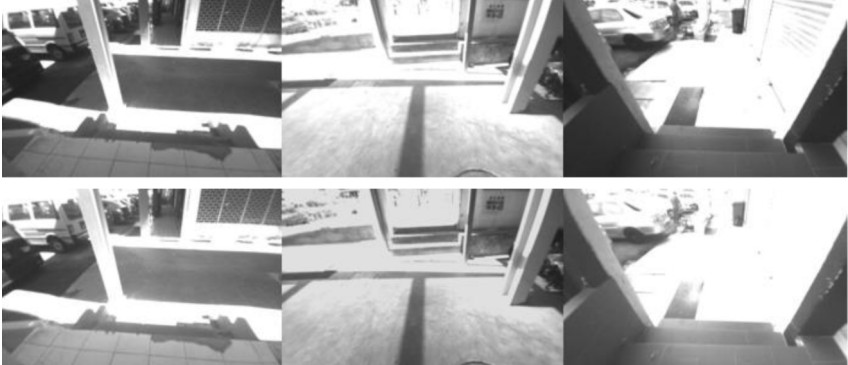

Figure 4: Three sample images before PCA whitening (top), and the three similar sample images after PCA whitening (bottom).

### 2.4.2 Generating the Depth Maps

The stereo camera used in the prototype has a baseline (distance between two camera lenses), *B* of 30.02 mm. Disparity (denoted as *x-x'*) is expressed in Equation (2), where *x* and *x'* are the distances between points in an image plane corresponding to the point on actual scene and their camera centres.

$f$ is the focal length of the camera, which is 2.0 to 2.1 mm. In other words, Equation (2) suggests that the depth of a point in a scene is inversely proportional to the displacement of corresponding image points and their camera centres. Since values of $B$ and $f$ are known as shown in Figure 5, the depth of all pixels in the image can be derived.

$$disparity = x - x` = Bf/z \qquad (2)$$

During data collection, lens undistortion and epipolar rectification of the images had been processed on real-time before saving them to the storage. Therefore, to compute the depth map, a two-step block matching technique (Table 2) can be used without concerning the lens distortion or epipolar rectification. Within these two steps, we paid attention to the regions of uniform intensity, smoothness of edges, and amount of noise on the depth maps by varying the edge detection filter, window size and disparity range. We observed the quality of the computed depth by visualizing them in coloured maps, before setting the best parameters to mass-produce all depth maps from the stereo image sequences.

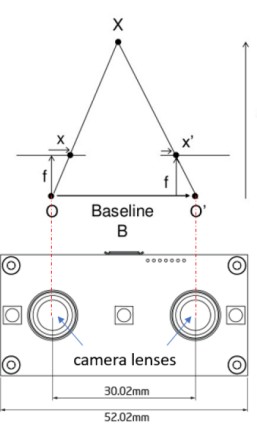

Figure 5: The equivalent triangles (top), and the dimensions of the stereo camera (bottom) used in the instrument.

**Regions of Uniform Intensity.** One of the biggest issues in most of the mapping outcomes is the occurrence of noise on large regions with very similar intensity. It appeared that when a large region has neighbouring pixels with uniform intensity, the mapping ended up with dusts of noise. This issue can be seen in the Figure 6. It is suspected that the intensity-based error minimization technique used in the process could have picked up very small differences and assigned some arbitrary disparity to that region. Since this is an issue relevant to intensity, it was left for error minimization mechanisms in machine learning to overcome it.

Table 2: A 2-step disparity mapping technique and its best settings used in this work.

| Step | Parameter |
|---|---|
| 1. Compute contrast based on edge detection. | Edge detection filter: Sobel |
| 2. Compute disparity with sum of absolute differences (SAD). | best window size: 9 
 best disparity ranges: (4, 20) |

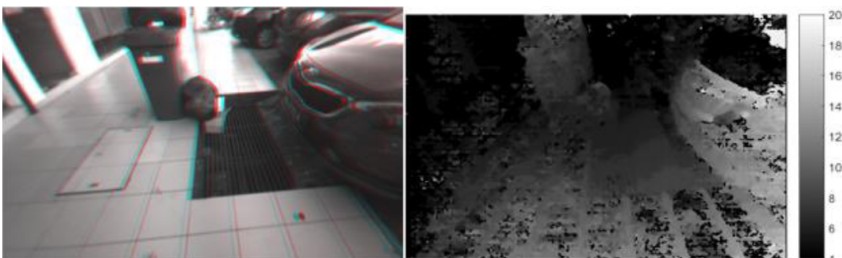

Figure 6: An issue of region with uniform intensity, in which the floor on the walkway (left) can be seen dusted with black dots or noise in the depth map (right).

**Windowing.** Window size has some significant effect on the disparity mapping. An optimal window size would make the map clean and clear. In several observations, when the window size was gradually increased, it helped to overcome noise better. However, when the window size was increased beyond an optimal value, it would produce some foggy and less accurate depth maps. It was found that a window size of 9 units could be optimal for most of the collected samples, although there were some exceptions, they were suspected to be just some small amount of outliers.

**Disparity Range.** Disparity range is the distance between the two cameras, and the distance between the cameras and the point of the actual scene. It was observed that this range must be optimized or else the mapping algorithm would get confused at the region where there is not too much of intensity variation. It was found that a range of (4, 20) is mostly acceptable for the samples. Using this range and the window size of 9, all 200 sets of depth maps were generated from the stereo image sequences.

### 2.4.3   Taxonomy of Surface Discontinuity and Data Annotation

The purpose of this work is to generate a dataset that can help develop assistive tool in negotiating surface discontinuity for the BLVs, and for that, the surface of a pathway must be distinguished between a "continuity" (indicating a smooth surface) and "discontinuity" (i.e. indicating a drop-off). Before annotating the data, a taxonomy of surface discontinuity was first developed. This taxonomy became an important guideline for the labelling task. We built the taxonomy based on the physical attributes of the collected data. It appeared to have 5 distinct classes of surface discontinuity namely (1) down-steps, (2) up-steps, (3) uncovered drainage, (4) drop-off without handrail, and (5) mixed gradient. We plotted the relative class frequency of the dataset in Figure 7. The classes are not quite balanced as both up- and down-steps have slightly higher frequency than the rest, which is a typical scenario in most of the urban areas we sampled the image sequences.

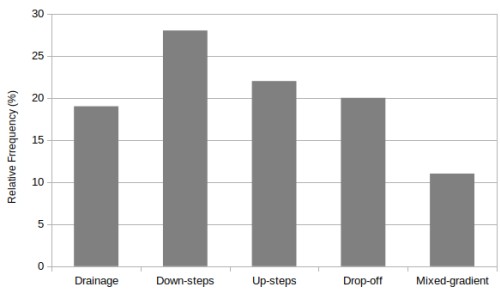

Figure 7: Relative frequency of the classes.

In addition to the physical attributes, the Uniform Building By-Laws Malaysia (1984) was also referred to guide the taxonomy development. For instance, in the case of steps (or staircases), the by-law 168 stated that: "the rise of any staircase shall be not more than 180 millimetres and the tread shall be not less than 255 millimetres and the dimensions of the rise and tread of the staircase so chosen shall be uniform and consistent throughout". Again for steps or staircases, the by-law 107 stated that: "Staircases exceeding 2225 millimetres in width shall be provided with intermediate handrail for each 2225 millimetres of required width spaced approximately equally". Our data contains some images of steps that violated these by-laws, and we classified and labelled them into up/down steps accordingly. We did the same for the other classes of data using this taxonomy.

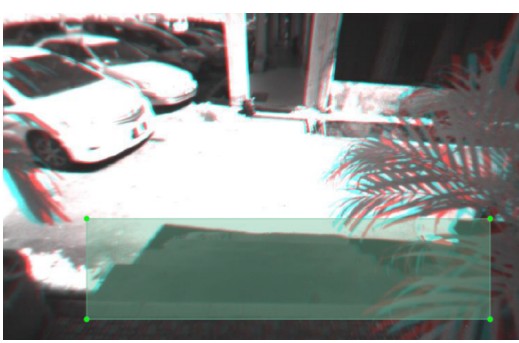 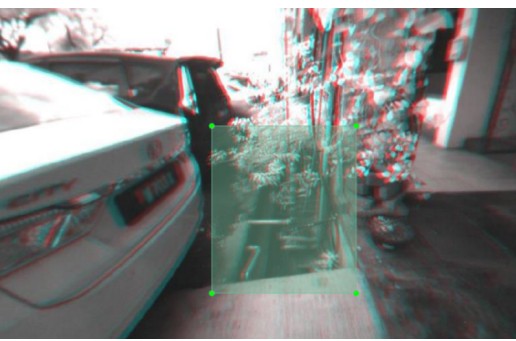

Figure 8: Examples of bounding box annotation on the anaglyph of two different samples - down-steps (left) and uncovered drainage (right).

During data annotation, we first mapped the image pairs into some red-cyan composite views (anaglyph), because it was challenging to visualize and make annotation on the depth maps directly. Then we annotated the bounding boxes of surface discontinuities using LabelImg. We generated the annotation in two file formats - PASCAL VOC XML file format, and COCO JSON file format. We originally collected 245 stereo image sequences, but by the end of all the pre-processing and annotation, we selected only 200 sequences of the best quality into the repository for publication. We

split the dataset into train/test sets by hand-picking 20 sets of very different image sequences as the test set (this contributed about 10%). Figure 8 shows examples of the bounding box annotation on the anaglyph of two different samples.

## 2.5 Potential Usages of the Dataset

With SurDis dataset that could exemplify the issue of surface discontinuity in urban outdoor environments, we propose the following usages:

- developing of wearable assistive tool that detects surface discontinuity in near real-time
- including of SurDis into other datasets that have various urban objects to train a more diverse object detection model targeting urban navigation
- utilizing of the depth map to extract distance information that can be supplied via the feedback mechanism of an assistive tool for blind navigation
- designing of evaluation system that rates the level of hazard for each class or each instance of surface discontinuity, this can become a hazard alert mechanism for an assistive tool in blind navigation

# 3 Experiments and Benchmark Models

We trained several benchmark models to provide baseline performance on this dataset. These models were trained on an NVIDIA GeForce GTX 1080 GPU with 16 GiB of RAM. We shared the code used for training the models at the same landing page for the dataset: https://github.com/kuanyewleong/surdis. We evaluated a few different model architectures ranging from an adapted Resnet-18 (Kaiming et al., 2016), Single Shot MultiBox Object Detector (SSD) (Liu et al., 2016), faster-RCNN (Shaoqing et al., 2015), to YOLOv4 (Alexey et al., 2020). We implemented the above models in the Torchvision framework, and leaving out any optimization or fine-tuning. Due to some minor class imbalance of the dataset, prior to model training we applied a random under-sampling technique by removing some samples of the majority class (mainly from down-steps and up-steps classes). We trained the models up to 200 epochs each. More details of the models used are described in Table 3, together with their corresponding evaluation results.

Table 3: Benchmark results for different models trained.

| Model | Backbone used | mAP |
| --- | --- | --- |
| Resnet | Resnet-18 | 0.156 |
| SSD | VGG-16 | 0.241 |
| Faster-RCNN | VGG-16 | 0.396 |
| YOLOv4 | Darknet-53 | 0.427 |

We evaluated all models by measuring their Mean Average Precision (mAP) on the test set. IOU threshold of 0.5 was applied in all of the mAP evaluation. Other parameters used in the training were documented on the given GitHub page. These preliminary experiments and results offer some idea of the quality of the dataset. We expect further development on the benchmark models with optimization on the hyper-parameters, augmentation, or some algorithmic improvement.

# 4 Limitations

There is still much to be done to expand this dataset to represent more diverse regions outside of the sampling locations as mentioned in Section 2.3. This dataset is limited to the Malaysian urban environments, and it might not have the rich diversity of surface condition of other nations.

The resulting depth map accuracy or the sensor depends on several factors i.e. (1) the algorithm used for extraction, (2) the frame rate and resolution, (3) illumination, (4) distance from sensor, and (5) camera calibration quality. We had applied the optimum configurations for item (2) to (5) for the sensor to work best during data collection, but we can't say the same for item (1). There are numerous

algorithms available for extracting depth map, and our choice of algorithm might not guarantee the most accuracy depth map. This is also the reason we included the bitmap stereo image pairs in SurDis for users to have some flexibility to re-generate the depth map using their choice of algorithms.

Another limitation is that since the data collection was performed based on a wearable setting, the height of the person conducting data collection might have some constraint on the samples.

## 5 Conclusion

We reviewed the existing research and datasets for blind navigation and realized some gaps in the works on surface discontinuity. With consultation from the BLV volunteers, we generated a localized SurDis dataset targeting the Malaysian urbans. This work offers a free and open dataset with the purpose of promoting the development of wearable assistive tool for blind navigation, especially in the low- and middle-income regions. We will maintain this work on the mentioned Github site that has a link to SurDis dataset on Zenodo. The use of Zenodo to host the dataset allows for versioning to ensure a consistent dataset usage. With the instrument and methodology used for the generation of SurDis documented, we invite contribution from researchers of similar interest to help expand this dataset. Researchers could also include this dataset by taking surface discontinuity as a new class label into currently available dataset to enrich the current models for blind navigation.

## Broader Impacts

Since the dataset could indirectly reflect the social and geo-political situation of the country in which basic infrastructures i.e. the walkways are potentially hazardous for the BLVs in some urban areas, it could potentially be utilized by certain parties to act upon the situation. The consequence might be positive if the involved stake holders are taking it constructively to improve the conditions of such infrastructures for the benefit of the BLVs. We do not foresee negative ethical consequences because of this work. This dataset would encourage more research and development of assistive tools for the benefit of the local BLV community, as well as the neighbouring regions who share similar built environments and conditions.

## Acknowledgements

Some major parts of this work including the preliminary investigation, development of a wearable instrument for data collection, 3D printing of the sample replicas for initial survey, and several other relevant research activities were conducted at Monash University. The facilities, technical support and other assistance provided by Monash University were greatly appreciated. We would also like to thank the three BLV service providers namely Dialogue in the Dark (Malaysia), St. Nicholas' Home Penang (Malaysia) and Vision Australia Dandenong (Victoria, Australia) for their valuable insights along different phases of this work. This research was partially funded by the Exploratory Research Grant Scheme from the Ministry of Education Malaysia, within the period of February 2015 to January 2016.

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
