# OpenReview forum: "SurDis: A Surface Discontinuity Dataset for Wearable Technology to Assist Blind Navigation in Urban Environments"
_NeurIPS.cc/2022/Track/Datasets_and_Benchmarks — NeurIPS 2022 Datasets and Benchmarks _

### Official Review · Reviewer_Vbjb · 2022-07-22
**New dataset but lack of sample codes.**

**Rating:** 7
**Confidence:** 4
**Correctness:** The dataset is constructed in a sound…

**Strengths:**

The dataset is ground-breaking in the blind and low vision problem solutions. It has the potential to boost such areas’ application development and research and to better serve people with visual impairments.

Judging from the data collection procedures, the dataset is well structured and source-reliable, It’s possible to broader the research community and arise studier's awareness of public service.

Also, the provided dataset could help in realistic wearable equipment designs.


**Weaknesses:**

The work mainly collected and introduced a pure dataset and its collection procedures. However, its real value on how exactly the data would help is not demonstrated, if the author could show a study case to reveal the data’s potential.

There is no example code for the usage of the dataset, if the author could demonstrate a piece of toy code, it will help users to get started easily.


**Additional Feedback:**

Please see my comments above.

**Clarity:**

The paper is written relatively clearly, but some minor typos should be examined. Such as: Typo problem at the beginning of paragraph 5, Introduction part “VIsual”. and the Reference first line, an extra line should be deleted.

**Documentation:**

The dataset was organized in a sound way, with details of each collecting step and purification procedures.
However, the maintenance plan is not clear, which might require more work to arrange.


**Ethics:**

No.

**Relation To Prior Work:**

The paper demonstrates related previous published work, with a clear explanation and introduced the proposed work's uniqueness.

**Summary And Contributions:**

The authors spared much effort to collect a set of Real path surface discontinuity data, aiming to cultivate and help researchers to assist the Blind and Low Vision people’s daily activities. The work is of good motivation and tends to solve real-world problems, which is worth encouragement.
In the data collecting process, the authors followed strict and rigorous procedures, from in-person consultation, and instrument development to data sampling, and afterward data quality analysis and preprocessing, making such resources reliable to support future work.
A dataset with long-term value usually requires a well-arranged maintenance plan, the author has done good collecting work, and some details for future management will be better. Besides, if the author could explore its usage in the real-world application using provided dataset would be better.

---

> ### Author Response · Authors · 2022-08-29
> **Revised version is made accordingly**
>
> We thanks the reviewers for their valuable time spent, and for offering very constructive comments to help us improve the paper’s quality.
>
> In response to Weakness:
> We accepted the suggestion and added some contents about experiment and baseline models.
>
> We actually did some intensive experiments and trained a range of baseline models prior to the submission of the manuscript. However, due to page limit and we considered to document more about the very unique data collection process, as well as to allow for more descriptions of the instrument developed for data collection, we thought of leaving the experiments or benchmark out. We have seen a few papers with a focus on dataset from NeurIPS 2021 or earlier without experiments or benchmark, for instance [1]. Thus it became our point of reference that we could trade-off the experiment part.
>
> However, after receiving similar comments from several reviewers, in which they pointed out the importance of sharing the experiment / benchmark / baseline models, we have now selected a few experiments and added them to the revised version. We have also updated our Github site with the code for model training and data loading, configurations to reproduce the results were documented there.
>
>
> In response to Clarity:
> Thanks for pointing out some typos, we will do extra proof-reading. Just to clarify, for the word “VIsual” at the beginning of paragraph 5 in Introduction section, it was meant to spell as such, because it is a special name given by the original author in [2]. The extra line in the first reference is now removed in our revised version.
>
>
> In response to Documentation:
> We have added some descriptions about the maintenance plan for the dataset and its landing page in the revised version. Thanks again for pointing out this important aspect.
>
>
> [1] https://datasets-benchmarks-proceedings.neurips.cc/paper/2021/hash/013d407166ec4fa56eb1e1f8cbe183b9-Abstract-round1.html
>
> [2] TETSUAKI B. 2021. VIDVIP: Dataset for Object Detection During Sidewalk Travel. J. Robotics Mechatronics. 375 33(5): 1135-1143.

---

### Official Review · Reviewer_CSu3 · 2022-07-24
**Review for SurDis**

**Rating:** 6
**Confidence:** 3

**Strengths:**

1 The paper is well motivated. In real world, identifying discontinuous surface areas is important to blind people. Previous works primarily focus on obstacles detection. The proposed SurDis can facilitate future research on discontinuous surface detection.

2 The images are collected from Malaysia and might be more challenging than images collected from developed countries.

3 The stereo pairs can provide valuable depth information about the scene.

**Weaknesses:**

1 It's beneficial to evaluate recent frameworks for object detection on SurDis. The examples in Fig. 9 seem to be easy. Thereby my major concern is the SurDis might be overwhelmed by existing frameworks even without depth information.

2 Selecting 10% of each sequence as test set might leak label information, because the images within a sequence are quite similar with each other.

3 In Sec 2.1, how many 3D replicas take part in the survey? (including the 3D replicas that are viewed as safe or easy by volunteers)





**Additional Feedback:**

It will be great to evaluate recent frameworks for object detection on SurDis.

**Clarity:**

This paper is well written and easy to understand.


**Correctness:**

Selecting 10% of each sequence as test set might leak label information, because the images within a sequence are quite similar with each other.

**Documentation:**

Some details are missing, for example:

1 In Sec 2.1, how many 3D replicas take part in the survey? (including the 3D replicas that are viewed as safe or easy by volunteers)

2 How many people are employed to take photos? It's important because the height of the person might have an impact on the view angle.

**Ethics:**

No, the author has anonymized the pedestrians, vehicles and property names.

**Relation To Prior Work:**

Related works have been discussed in detail.

**Summary And Contributions:**

The paper introduces a novel dataset that focuses on detecting surface discontinuities to help blind people navigate in urban areas. The dataset is collected from 200 sets of stereo image sequences from 10 different locations in Klang Valley, Malaysia.

---

> ### Author Response · Authors · 2022-08-29
> **Revised version is made accordingly**
>
> We thanks the reviewers for their valuable time spent, and for offering very constructive comments to help us improve the paper’s quality.
>
> In response to Weakness (1) and Additional Feedback:
> We accepted the suggestion and added some contents about experiment and evaluation. Fig 9 might not reflect the actual complexity of the issue, quite some scenes from our dataset have multiple overlapping objects. However, in the related paragraph to Fig 9, we were trying to introduce the idea of annotating on the sample’s anaglyph, thus we inserted some simplest annotated images just to depict the idea.
>
> We actually did some intensive experiments and trained a range of baseline models prior to the submission of the manuscript. However, due to page limit and we considered to document more about the very unique data collection process, as well as to allow for more descriptions of the instrument developed for data collection, we thought of leaving the experiments or benchmark out. We have seen a few papers with a focus on dataset from NeurIPS 2021 or earlier without experiments or benchmark, for instance [1]. Thus it became our point of reference that we could trade-off the experiment part.
>
> After receiving similar comments from several reviewers, in which they pointed out the importance of sharing the experiment / benchmark / baseline models, we have now selected a few experiments and added them to the revised version. We have also updated our Github site with the code for model training and data loading, configurations to reproduce the results were documented there.
>
>
> In response to Weakness (2) and Correctness:
> Thanks for pointing out the issue about the test set, this is really our big mistake in the writing. Our researchers had prepared the test set from several different sequence sets representing about 10% of the total samples, instead of using 10% from each sequence as per said in the first submission. It was basically a miscommunication in our writing part. We have amended the contents in regards to the test set in the revised version.
>
>
> In response to Weakness (3) and Documentation (1):
> Taking the suggestion from the reviewer, we have included more details about the 3D models used in the pre-collection stage in the revised version of our manuscript. We added details of the amount of 3D models involved in the survey, including the 3D replicas that are viewed as safe or easy by volunteers.
>
>
> In response to Documentation (2):
> To make sure that our samples are consistent from the point of the sensor placement, we only used one single person for all the data collection tasks. We documented several measurements of our sensor relative to its surrounding i.e. the height of the sensor to the ground surface when it is put on as a wearable device by our researcher, the forward facing angle of the sensor, the distance from the researcher to the focal point of the sensor and so on. Please refer Section 2.2 in our original submission.
>
>
>
> [1] https://datasets-benchmarks-proceedings.neurips.cc/paper/2021/hash/013d407166ec4fa56eb1e1f8cbe183b9-Abstract-round1.html

---

### Official Review · Reviewer_S4AE · 2022-07-26
**A surface discontinuity dataset for navigation in urban environments with a wearable technology**

**Rating:** 7
**Confidence:** 4
**Clarity:** The quality of the paper is acceptabl…

**Strengths:**

The creation of mobility tools for BLV people seldom considers problems with surfaces, assuming their continuity. The set of depth maps from this collection would be helpful to address this issue if the surface discontinuity poses a serious problem in the area of application. Instead of using typical hardware, the authors build a wearable sensor and used it to capture the data.

**Weaknesses:**

The number of works in the literature on the subject of surface discontinuity for wearable sensors for BLV people is limited, confining the number of researchers interested in the usage of the dataset. The mentioned in the paper way of integrating the dataset with other datasets is not shown and requires further investigation on the possible problems with implementation. The dataset is collected in a specific environment which for many application domains would not be relevant (buildings, even pavements) or is not representative enough (country road, a footpath in the park). It is not certain if applications developed with depth maps created with the specific wearable sensor would work with other depth sensors.

**Additional Feedback:**

1.	It is not clear how printed 3D replicas would support potential users of the dataset.
2.	The limitations imposed by the wearable sensors should be discussed. Researchers would create their own sensors and datasets instead of using a dataset and have problems with sensor construction with parts that may not be available in the nearest future.
3.	The integration of different sensors or depth maps, as mentioned, should be investigated. What would the extension of the dataset look like? Readers are encouraged to do it but would they know how? Are the depth maps from different sources easily merged or used together in the same software?
4.	Are the selected ten locations enough to cover the variability of the surfaces in urban environments? How do we know they are representative?
5.	A performance evaluation of a typical machine learning method on this dataset would be a nice addition to show the usability and difficulty of the dataset.
6.	At https://github.com/kuanyewleong/surdis it is written that “it is reccommended that when you are training your model using this dataset, you may slightly expand the bounding boxes (i.e. add a few units of pixel to all edges).” This would make the repeatability of the results for different researchers more difficult. Why was the suggested addition not performed?


**Correctness:**

The submission seems correct. However, it lacks information on the diversity of classes. Is the dataset balanced? Also, there are only five classes. In my opinion, examples of flat surfaces or surfaces assessed by a specialist as non-hazardous should be present in the dataset. If a “normal” case (the sixth class) is not present, a surface classifier will always assign one of those classes to input depth maps. This requires clarification and explanation. Furthermore, it should be explicitly ensured in the text that depth maps from testing and training subsets are disjoint.



**Documentation:**

The documentation is not sufficient. The paper does not contain information on the class distribution. The dataset organization and remaining information is in sufficient detail.

**Ethics:**

There are no ethical concerns that warrant further discussion or review.

**Relation To Prior Work:**

The dataset is compared with data collections designed for a similar purpose.

**Summary And Contributions:**

The main contribution of this work is a collection of labeled depth maps of 10 locations that contain different surfaces. It is the first dataset that considers surface discontinuity while creating mobility tools for blind or low vision (BLV) people.

---

> ### Author Response · Authors · 2022-08-29
> **Revised version is made accordingly (part 1)**
>
> We thanks the reviewers for their valuable time spent, and for offering very constructive comments to help us improve the paper’s quality.
>
> First and foremost, we want to take the opportunity here to make it clear that this work was conducted based on a design science research approach. Firstly, we were informed by the BLV service providers about the surface discontinuity issue in Malaysia, specifically in the urban areas. Secondly, along the development of the data collection instrument, pre data collection and during data collection phases, we followed the said design science research process to continuously engage in consultation / feedback with our collaborated BLV service providers. This process helped to make sure that the samples we collected are specific to the BLVs navigation issue, and the wearable instrument is acceptable by the BLV people (including its unobtrusive nature, its size, its weight, its configuration and setup around an BLV person).
>
> Knowing our research approach could help address some of the comments / doubts raised by the reviewer in several of our responses below. Due to page limit, we excluded our design science research method in the manuscript.
>
> In response to reviewer’s comments:
> Reviewer: “The number of works in the literature on the subject of surface discontinuity for wearable sensors for BLV people is limited, confining the number of researchers interested in the usage of the dataset. ”
>
> Our response:
> It is due to the lack of surface discontinuity research / datasets, that the literature available for reference is limited. Through consultation with the local BLV service providers, we found this research gap and supplemented it with our work. Based on WHO [1], the world wide distribution of BLV people in low- and middle-income regions is estimated to be four times higher than in high-income regions [2]. From our literature review (please refer the Introduction section of our manuscript), we found that most of the technologies / research works related to BLV navigation were conducted in high-income countries / cities, or by researchers from high-income regions, and they have some very different range of focuses. This is an obvious research gap. Topping with the fact that surface discontinuity issue is less apparent in developed cities (and their barrier-free access facilities at public walkways are mostly well-equipped), it could explain why machine learning communities from high-income regions placed limited attention to such issue that largely affects the local BLVs within the low- and middle-income regions. Research funds in low- and middle income countries were mainly spent on addressing issues that are related to combating diseases (these regions account for ~85% of the global disease burden ), poverty-related malnutrition, famine, agriculture (to tackle hunger), water scarcity and etc [5]. Little is done to offer improvement for the BLV communities as policy makers could be viewing the above issues as more critical. All of the aforementioned factors could limit the number of works in the literature on the subject of surface discontinuity for wearable sensors for BLV people.
>
> We look at our work in a positive way that even if the limited literature might confine the number of researchers’ interest in the usage of the dataset, we are still very much keen to publish this dataset. We believe someone has to make this issue aware to a wider audience / researchers. And we are trying to contribute to a real world problem here (that affects about 2.2 billion people in predominantly middle to low income regions), via a platform that has a strong base of machine learning / IT researchers like NeurIPS.

---

> > ### Author Response · Authors · 2022-08-29
> > **Revised version is made accordingly (part 2)**
> >
> > Reviewer: “The mentioned in the paper way of integrating the dataset with other datasets is not shown and requires further investigation on the possible problems with implementation. ”
> >
> > Our response:
> > This statement about “integrating the dataset with other datasets ” was mentioned in our Conclusion section, suggesting some possible future application in a bigger scope, or for a region that has a different need for blind navigation. For example, one can utilize some other datasets for pedestrians, or vehicles, and complement them with our surface discontinuity dataset to train a detection model that cater for more classes of objects. This can be easily done since our data annotation is readily prepared in the commonly used COCO and PASCAL VOC formats. Furthermore, we have provided the original raw image pairs, a part from the extracted depth maps (which are also in standard numpy array format). We mentioned about these standard formats of our work in the original submission. This should be sufficient enough for anyone within this field to work on the integration. We do not provide further details of how these datasets can be integrated. We see that this has not direct connection to the publication of our dataset, it is just our suggestion of future potentials, thus we leave it to the future research works.
> >
> > Reviewer: “The dataset is collected in a specific environment which for many application domains would not be relevant (buildings, even pavements) or is not representative enough (country road, a footpath in the park).”
> >
> > Our response:
> > This dataset is never intended for many application domain. We stated this clearly in our manuscript, and our title has reflected that it is specific for wearable technology to assist the BLVs to negotiate surface discontinuity.
> >
> > We have based on our consultation with the BLV service providers to identify the 5 most hazardous classes in blind navigation around the urban. We were told by the BLV volunteers that if they were to adopt some technology to assist them in navigation, they will not remove their traditional guide cane. The adoption of navigation technology is not as pervasive as expected [3]. And for the issues or objects that can be easily addressed by the guide cane (i.e. objects above surface, walls, smooth surface, and most built structures that complies to standard regulations), they do not prefer intervention by technology – they rather apply their guide cane.
> >
> > Thus, with the help of the BLV volunteers, we eventually identified the 5 classes of surface discontinuity that are hardly address by the guide cane, and that are highly hazardous to BLVs during navigation. Hence samples like country road, a footpath in the park etc were not included, not only because they can be handled by the guide cane, these places are not essential locations for a BLV person to travel to. Excluding non-essential classes in developing assisting tool for the BLVs is crucial, this is because blind navigation is a very challenging task and it requires even a well-trained BLV person to put in a lot of attention (before a BLV person is able to use a guide cane for navigation, he/she has to go through some “Orientation and Mobility” training by a certified trainer). When developing such an assistive tool, we want to reduce as much as possible the interruption from technology, align it to the usage of a guide cane, and help detect only those identified classes of hazards.
> >
> > We want to point out again (which we already stated in our original submission), that through our findings, not all surface discontinuities are troublesome for blind navigation. For instances, properly covered drainage, steps with even rises, rises of steps built according to local regulations, and drop-off with hand-rail are not hazardous to BLV people.
> >
> > Due to page limit, the many insights (such as all of the above) that we gained from our collaboration with the BLV service providers were not included in the manuscript. As such, we want to assure our reviewer that we have strong justification for the selection of the 5 classes, and not the other excluded surface types.

---

> > > ### Author Response · Authors · 2022-08-29
> > > **Revised version is made accordingly (part 3)**
> > >
> > > We have also clarified the dataset limitation in our original submission. Surface discontinuity faced by the BLVs could be a global issue that requires localized dataset due to the nature that built structures around the urban environment could be very different across the world (or even within a nation). Such issue might be less apparent in well-developed cities, while it could be more common in developing / under-developed cities. Once the dataset is published, we can tap on global talents to contribute their technological / machine learning expertise to help address such a local issue.
> > >
> > > Additionally, we also stated in the introduction that the presence of universal access facilities such as tactile ground surface indicators, handrails along staircases, pedestrian ramps, properly covered drainage and etc differ from one city to another. Cities with better universal accessibility tend to have less issues with surface discontinuity for the BLVs, and vice-versa. This factor will affect the samples diversity and size that could be collected from a particular location.
> > >
> > > To conclude, we want to highlight to the research communities that the above insights reveal that the issue of surface discontinuity for BLVs’ navigation requires local datasets and local machine learning models. It is not uncommon to see more and more contributions of local datasets from the machine learning communities, an example from NeurIPS is [2] published in 2019.
> > >
> > >
> > > Reviewer: “It is not certain if applications developed with depth maps created with the specific wearable sensor would work with other depth sensors.”
> > >
> > > Our response:
> > > We have revealed all information / details about our sensor, the configurations used, and the algorithm and its parameters applied for depth maps generation (either in the manuscript, Appendix or on the Github page). We know that as long as the sensor specifications are not too different from ours, and one apply the similar configuration / setting, there should be little compatibility issues.
> > >
> > > In response to Correctness:
> > > We described in brief how the 5 classes were discovered from the survey with the BLVs using 3D models. We also referred to the country regulations (by-laws) to help decide urban surfaces that could be hazardous, and to include such samples in the dataset. Thanks to the comments from the reviewer, we have now added more details about the selection of these 5 classes. We also included the frequency / distribution of them with some additional details in the revised version.
> > >
> > > About the “normal” class as per suggested by reviewer, they are regarded as the background or other none hazardous feature on a pathway. We have referred to several popular object detection datasets i.e. Cifar, COCO and ImageNet, and found that the “normal” or background class is not annotated. We understand that most modern machine learning tool-kits will treat whatever we don't label / annotate as background and they will become an implicit class which is typically labeled 0 or -1 by default. Generally, in most of these tool-kits there are facilities to crop and resize any random patches from an image aside of those annotated classes as background during model training. A typical practice is to extract patches that have low "intersection over union" with any object present in the image frame from the annotated classes.
> > >
> > > On the comment about “it should be explicitly ensured in the text that depth maps from testing and training subsets are disjoint”. We did describe in our original submission that we split the testing set and training set. Please refer our original version at line 284 to 285. In our latest revised version, we made a small update to this info with more details.

---

> > > > ### Author Response · Authors · 2022-08-29
> > > > **Revised version is made accordingly (part 4)**
> > > >
> > > > In response to Additional Feedback:
> > > > 1. It is not clear how printed 3D replicas would support potential users of the dataset.
> > > >
> > > > We mentioned clearly the usage of 3D replicas in our original submission that they were used for survey between our researchers and the BLV volunteers, in order to understand the type of surface discontinuity that are hazardous to their navigation, such that we can specifically target those types of surface discontinuity during data collection. (please refer line 113 to 119 in our original submission).
> > > >
> > > > Since BLV people can’t see (or can’t see well), we were not able to present videos/photos of the potential samples during our consultation, and we were not allowed to bring them out to the field for further examination due to risk and ethical concerns. Hence, 3D replicas were the best option we had, for them to touch and feel using their fingers, and response to our survey.
> > > >
> > > > These 3D replicas are never intended for the potential users of the dataset (we never mentioned this in our original submission). We have contributed a novel idea of how future researchers from other regions can work on such issue using 3D printing, especially during pre-data collection survey with the BLV people.
> > > >
> > > >
> > > > 2. The limitations imposed by the wearable sensors should be discussed. Researchers would create their own sensors and datasets instead of using a dataset and have problems with sensor construction with parts that may not be available in the nearest future.
> > > >
> > > > We provided a brief discussion about the sensor limitations. We think that the limitation of the wearable sensor is not closely relevant to our purpose of this paper, and we have limited pages for other more relevant topics. More importanly, our intention here is to release to the public the readily usable dataset we have collected and pre-processed. Analysis on the limitations of the wearable sensor is a big topic in this project. Indeed we have done analyzing its limitations, and it involves several aspects of power consumption, HCI design, sensitivity, algorithmic efficiency bounded by the sensor, weather conditions, lighting, BLV users’ body gestures during navigation and several others, the page limit of NeurIPS won’t allow such discussion. We agreed that discussion about the dataset’s limitations is definitely critical (which were provided), but discussion about the sensor limitations seems to be deviated from our main intention of publishing the dataset.
> > > >
> > > >
> > > > 3. The integration of different sensors or depth maps, as mentioned, should be investigated. What would the extension of the dataset look like? Readers are encouraged to do it but would they know how? Are the depth maps from different sources easily merged or used together in the same software?
> > > >
> > > > One of our responses above provides explanation to these questions, please refer some earlier section above. We need to point out that, depth maps are consisting of some arrays of numerical values, the usage of them are not bound to software even if depth maps from several sources are used together. And we have provided the most widely used numpy array format in our work. We have also provided the complete details of configuration and setting of the sensor in the Appendix. This is akin to how machine learning researchers have no problem using images taken by different cameras, or videos recorded by different recorders, or sound datasets captured by different microphones.
> > > >
> > > > Pondering through the above comment, we asked ourselves the following questions:
> > > > (a) Is there a global standard (or a de facto) for sensor choice in depth maps generation?
> > > > (b) Did the sensor choice/type of all the past datasets of depth maps affect their acceptance in NeurIPS publications (if there is not global standard for sensor choice as for now)?
> > > > (c) Did all contributors of popular image datasets i.e. COCO, ImageNet, CIFAR use similar type of digital camera?
> > > > (d) Were images from the like of COCO, ImageNet, and CIFAR datasets observed to encounter integration or merging issues by the research communities?
> > > >
> > > > What we knew so far is, it appears to be a “no” to all the questions above. As such, we have a strong ground to believe that the sensor choice and the integration issue should not raise much concerns.

---

> > > > > ### Author Response · Authors · 2022-08-29
> > > > > **Revised version is made accordingly (part 5)**
> > > > >
> > > > > 4. Are the selected ten locations enough to cover the variability of the surfaces in urban environments? How do we know they are representative?
> > > > >
> > > > > We have elaborated in the beginning about our design science research approach involving the BLV service providers, which had helped identify not only the types of hazardous surface discontinuity, but also pointed out to us several essential locations with such issue that affects their daily functions. These are essential locations that offer social, public and business functions to the BLVs.
> > > > >
> > > > > Secondly, not every location or every city in the region required assistive tool for the BLVs to navigate. Some locations/cities are much more modern, and were developed according to strict building regulations, and some even offer quality universal access facilities. The ten locations we had selected are not only lack in such facilities in most of the surrounding, but a number of the built structures were substandard and not built according to the Uniform Building By-Laws [4]. These locations were also observed to have high frequency of the identified classes of surface discontinuities. We can’t deny that some other locations might have issue of surface discontinuity too, but our pre data collection survey found that they were very low in frequency, thus the samples we can collect from such low frequency locations might be very limited. Due to manpower and resource constraints, we focused our sampling locations at the said ten locations.
> > > > >
> > > > > Go back to the reviewer’s comment, how do we know they are representative? We have now included some experiments and baseline models in the revised version. From the preliminary results, we can deduce that the dataset can support machine learning for the targeted region. There are of course rooms of improvement. Despite that, we remain cautious that the dataset is diverse and large enough for some global applications. We have clarified this in the limitation section.
> > > > >
> > > > >
> > > > > 5. A performance evaluation of a typical machine learning method on this dataset would be a nice addition to show the usability and difficulty of the dataset.
> > > > >
> > > > > We accepted the suggestion and added some contents about experiment and performance evaluation in the revised version.
> > > > >
> > > > >
> > > > > 6. At https://github.com/kuanyewleong/surdis it is written that “it is recommended that when you are training your model using this dataset, you may slightly expand the bounding boxes (i.e. add a few units of pixel to all edges).” This would make the repeatability of the results for different researchers more difficult. Why was the suggested addition not performed?
> > > > >
> > > > > We have performed the suggested addition, thanks for the comment. Our initial thought was to provide the original annotation of this datasset to the users, and it will be up to the users to apply any techniques  i.e. expand the bounding box (they may expand it according to their analysis), apply various data augmentation, crop the samples into square boxes to suit most HOG-based SVM models and etc. After all, the users should have the freedom to apply various techniques to further pre-process the data, as a mean of improving their detection models (if needed). The very reason for the original annotation to be tightly performed was that it served as a measure for quality control. Since different human annotators might have biases when annotating the bounding boxes, it is a common practice that objects are annotated tightly around their borders. This resulted in better quality of annotation.
> > > > >
> > > > >
> > > > >
> > > > > References:
> > > > >
> > > > > [1] https://www.who.int/news-room/fact-sheets/detail/blindness-and-visual-impairment
> > > > >
> > > > > [2] Vision Loss Expert Group of the Global Burden of Disease Study. Causes of blindness and vision impairment in 2020 and trends over 30 years: evaluating the prevalence of avoidable blindness in relation to “VISION 2020: the Right to Sight”. Lancet Global Health 2020. doi.org/10.1016/S2214-109X(20)30489-7
> > > > >
> > > > > [3] GIUDICE, N. A. & LEGGE, G. E. 2008. Blind Navigation and the Role of Technology. In: HELAL, A., MOKHTARI, M. & ABDULRAZAK, B. (eds.) Engineering Handbook of Smart Technology for Aging, Disability and Independence. John Willey & Sons.
> > > > >
> > > > > [4] Street, Drainage and Building Act 1974 (Act 133), Uniform Building By-Laws 1984. National
> > > > > 331 Council of Local Government, Malaysia.
> > > > >
> > > > > [5] https://www.frontiersin.org/articles/10.3389/frma.2019.00003/full

---

> > > > > > ### Comment · Reviewer_S4AE · 2022-08-29
> > > > > > **The revision improved the quality of the work**
> > > > > >
> > > > > > The revision and provided discussion clarified most issues. I've updated my score for the paper.

---

### Official Review · Reviewer_BwNo · 2022-07-27
**This work tackles an important problem but it is hard to gauge how useful the dataset is without any experiments.**

**Rating:** 7
**Confidence:** 4

**Strengths:**

This paper has several strengths, and I appreciate the problem that it is trying to address. Blind navigation is a difficult problem, especially in less-developed urban environments.
-it is clear the technology and collection platform was developed in collaboration with blind people and experts in the area. The chest-mounted device can be used in conjunction with existing navigation apparatus (e.g., cane). The researchers also consulted O&M experts and used 3D tactile models to inform data collection with blind people.
-as the authors state, the dataset is collected in a unique environment where accessibility challenges are perhaps more widespread (urban Malaysia). A model or technique that succeeds under these conditions could perhaps be more useful in practice.
-the dataset's focus on depth map sequences as its primary modality could support diverse areas of ML research: traditional vision-based approaches (e.g., image/video object detection/segmentation) as well as potentially point-cloud models.


**Weaknesses:**

My main concern with this work is that no experiments were performed to validate the usefulness of the dataset. While the focus of the paper is the dataset itself, I would like to see some evidence (e.g., preliminary/baseline models that suggest it's possible to detect surface discontinuities using this dataset). Otherwise, it is unclear if the data is too "noisy" to be useful for ML approaches. I believe it is possible to address these concerns during the rebuttal period and I am open to raising my score if they are.
-lacking empirical experiments: the paper lacks any preliminary experiments that show the data is useful for learning detection models. Such experiments would also be useful for delineating the set of tasks that this dataset could be used to support. A simple object detector model trained on the depth maps could be a good start, since the annotations are already in the COCO format.
-missing some details: while some parts of the data collection were thorough, others were not. for example, how many 3D models were used in the pre-collection stage with blind participants? Of these, how were the eventual 5 classes chosen? It was also difficult to find info about the relative frequency of these 5 classes throughout the dataset.
-minor writing problems: the paper contains some minor writing errors that could easily be fixed with an automated grammar checker.


**Additional Feedback:**

* Post rebuttal comments:
The authors have thoroughly addressed my initial concerns and I believe the paper stands much stronger now. I would argue for its acceptance.

**Clarity:**

besides a few minor writing errors, the paper was easy to follow. Some details are missing and should be included (see weaknesses).


**Correctness:**

As I mentioned earlier in my review, I appreciate the collection procedure where the authors consulted with blind people and experts in the field. That leads me to believe that the types of scenarios, discontinuity types, and collection apparatus are ecologically valid. However, I believe more evidence is needed to show that the dataset is diverse, large, and clean enough to support machine learning in this space.


**Documentation:**

the paper thoroughly documents the collection process and apparatus. The github link includes the relevant code for the collection hardware and processing scripts. I believe the repository could be improved with additional documentation and example code on how to load samples from the dataset (e.g., a jupyter notebook).

**Ethics:**

I do not see any ethical problems with this dataset.

**Relation To Prior Work:**

The paper does discuss its relation to prior work, namely other datasets used for blind navigation. Differences include the collection environment and the unique collection apparatus that was built for this data collection. One tangentially related work that I happen to know of in this area (Project Sidewalk (https://sidewalk-sea.cs.washington.edu)) was not and probably should be included.


**Summary And Contributions:**

In this paper, the authors introduce SurDis, a dataset of surface discontinuities captured by a chest-mounted depth camera. The dataset and models trained using it could be useful for improving navigation for blind and visually-impaired people by automatically detecting discontinuities (hazards) in front of them. In contrast to other relevant datasets, SurDis was collected in a unique urban environment where such surface discontinuities are more prevalent.

---

> ### Author Response · Authors · 2022-08-29
> **Revised version is made accordingly**
>
> We thanks the reviewers for their valuable time spent, and for offering very constructive comments to help us improve the paper’s quality.
>
> In response to Weakness:
>
> About baseline models / experiments to validate the dataset:
> We accepted the suggestion and added some contents about experiment and baseline models.
>
> We actually did some intensive experiments and trained a range of baseline models prior to the submission of the manuscript. However, due to page limit and we considered to document more about the very unique data collection process, as well as to allow for more descriptions of the instrument developed for data collection, we thought of leaving the experiments or benchmark out. We have seen a few papers with a focus on dataset from NeurIPS 2021 or earlier without experiments or benchmark, for instance [1]. Thus it became our point of reference that we could trade-off the experiment part.
>
> However, after receiving similar comments from several reviewers, in which they pointed out the importance of sharing the experiment / benchmark / baseline models, we have now selected a few experiments and added them to the revised version. We have also updated our Github site with the code for model training and data loading, configurations to reproduce the results were documented there.
>
> About 3D models used in the pre-collection stage:
> Taking the suggestion from the reviewer, we have included more details about the 3D models used in the pre-collection stage in the revised version of our manuscript. We added details of the amount of 3D models involved in the survey, how was the survey conducted between the BLVs and the interviewers, and more descriptions about what types of surfaces were considered safe/hazardous based on the BLVs using the 3D models.
>
> Taxonomy of the 5 classes:
> We described in brief how the 5 classes were discovered from the survey with the BLVs using 3D models. We also referred to the country regulations (by-laws) to help decide urban surfaces that could be hazardous, and to include such samples in the dataset. Thanks to the comments from the reviewer, we have now added more details about the selection of these 5 classes. We also included the frequency of them with some additional details in the revised version.
>
>
> In response to Correctness:
> As mentioned above, we have now included some experiments and baseline models in the revised version. From the preliminary results, we can deduce that the dataset can support machine learning for the targeted Klang Valley. There are of course rooms of improvement. Despite that, we remain cautious that the dataset is diverse and large enough for some global applications. We have clarified this in the limitation section. Surface discontinuity faced by the BLVs could be a global issue that requires local dataset due to the nature that built structures around the urban environment could be very different across the world (or even within a nation). Such issue might be less apparent in well-developed cities, while it could be more common in developing / under-developed cities.
>
> Secondly, we also stated in the introduction that the presence of universal access facilities such as tactile ground surface indicators, handrails along staircases, pedestrian ramps, properly covered drainage and etc differ from one city to another. Cities with better universal accessibility tend to have less issues with surface discontinuity for the BLVs, and vice-versa. This factor will affect the samples diversity and size that could be collected from a particular location.
>
> Taking this opportunity, we want to highlight to the research communities that the above insights reveal that the issue of surface discontinuity for BLVs’ navigation requires local datasets and local machine learning models. It is not uncommon to see more and more contributions of local datasets from the machine learning communities, an example from NeurIPS is [2] published in 2019.
>
>
> In response to Relation To Prior Work:
> Thanks for pointing out Project Sidewalk, we have included it in the review of prior works section. It serves as a good example of how we can expand the dataset in the future by attracting citizen scientists to contribute in data discovery and annotation.
>
>
> In response to Documentation:
> We have updated our Github with tutorial code for data loading, as well as some scripts for models training and evaluation, together with their configuration.
>
>
>
>
> [1] https://datasets-benchmarks-proceedings.neurips.cc/paper/2021/hash/013d407166ec4fa56eb1e1f8cbe183b9-Abstract-round1.html
>
> [2] https://proceedings.neurips.cc/paper/2019/hash/ee389847678a3a9d1ce9e4ca69200d06-Abstract.html

---

### Official Review · Reviewer_HYwb · 2022-07-28
**A great dataset for assistive technology**

**Rating:** 7
**Confidence:** 3
**Correctness:** The claims are correct to my knowledge
**Clarity:** The paper is well written

**Strengths:**

 - The authors design the dataset around feedback from potential users of the assistive technology. Using 3d printed models as aids for identifying the appropriate discontinuities to target was an excellent addition.
- The specific data collection procedure is outlined in detail
- The data is permissively licensed, easily accessible, and well documented


**Weaknesses:**

- The authors did not implement a baseline model to evaluate the usefulness of the dataset as the training set for an ML application.
- The paper does not evaluate how sensitive an assistive system would be to the camera configuration and environment, although the authors do acknowledge that this is a limitation


**Additional Feedback:**

I think this is a great dataset. I would suggest the authors create a baseline model, a well-defined evaluation procedure, and a leaderboard in order to encourage others to use the data.

The authors could also enlist the help of the community to collect more data in their area using the same prototype data collection tool. Crowd sourcing the next iteration of the dataset seems critical to expanding its size, which is necessary for future adoption.

**Documentation:**

The dataset is well documented

**Ethics:**

The ethical concerns are discussed

**Relation To Prior Work:**

Prior work is discussed in the intro and the authors do a good job explaining how their dataset fills a gap in the existing work

**Summary And Contributions:**

The paper presents a dataset of labeled surface discontinuities in depth maps and stereo images. The dataset can be used to create assistive technology for blind and low vision people. The authors provide substantial detail on the specific data collection procedure and artifacts of the dataset.

---

> ### Author Response · Authors · 2022-08-29
> **Revised version is made accordingly**
>
> We thanks the reviewers for their valuable time spent, and for offering very constructive comments to help us improve the paper’s quality.
>
> In response to Weakness:
> Point 1:
> We accepted the suggestion and added some contents about experiment and baseline models.
>
> We actually did some intensive experiments and trained a range of baseline models prior to the submission of the manuscript. However, due to page limit and we considered to document more about the very unique data collection process, as well as to allow for more descriptions of the instrument developed for data collection, we thought of leaving the experiments or benchmark out. We have seen a few papers with a focus on dataset from NeurIPS 2021 or earlier without experiments or benchmark, for instance [1]. Thus it became our point of reference that we could trade-off the experiment part.
>
> However, after receiving similar comments from several reviewers, in which they pointed out the importance of sharing the experiment / benchmark / baseline models, we have now selected a few experiments and added them to the revised version. We have also updated our Github site with the code for model training and data loading, configurations to reproduce the results were documented there.
>
> Point 2:
>
> We did not evaluate how sensitive an assistive system would be to the camera configuration and environment, and we acknowledged in the last section of our manuscript that this is a limitation in our work for the meantime. In spite of that, we provided the stereo camera sensor’s details in the appendix (i.e. model, manufacture, camera resolution, stereo baseline length, average distance of the focal point of the camera to the surface, field of view and so on) and configuration used in this dataset, these details should allow users to derive the sensitivity of the assistive system.
>
> Additionally, such evaluation will be more meaningful and reliable if we may conduct it with some volunteers from the blind and low vision (BLV) communities, instead of using some sighted volunteers. This is largely due to 2 main reasons: (1) BLV people walk in a very different pace as compared to sighted people, (2) BLV people walk with a guide cane tapping (or swinging for quicker pace) on the surfaces repeatedly to the left and right. In order to conduct such evaluation, we will have to go through a different ethics consent with the NGOs that we are working with. This will be our next phase of the research looking at the hurdle / ethical clearance needed.
>
>
> In response to Additional Feedback:
> Put forward a site to describe the evaluation procedure with a leaderboard is a viable suggestion. We are seeking extra resources/manpower to do so in our next phase of this project.
>
> We welcome the idea for crowd sourcing to expand the dataset. We had in fact initiated this project since late 2016 with crowd sourcing on some social media platforms such as this site: https://www.facebook.com/Assistive-Technologies-265305997151982. However, they didn’t receive much responses, thus we changed our strategy to collect the data ourselves. We still believe crowd sourcing is our next iteration approach to expand this dataset, if execute with the right strategy. Thank you for the feedback.
>
>
>
> [1] https://datasets-benchmarks-proceedings.neurips.cc/paper/2021/hash/013d407166ec4fa56eb1e1f8cbe183b9-Abstract-round1.html

---

### Official Review · Reviewer_TD3P · 2022-07-28
**An interesting social support to BLV community.**

**Rating:** 6
**Confidence:** 4
**Correctness:** The dataset is constructed and annota…

**Strengths:**

The paper and the presented dataset address an important topic to society of assistive technology. It shows superiority over the cited existing datasets. The dataset is accessible via Github and well explained.   The dataset has claimed several potential usages like developing wearable assistive devices, integrating with other datasets, and evaluating the level of hazardous. The reviewer agrees on all of them.

**Weaknesses:**

The main weakness of the presented work is that it does not show a comprehensive analysis and experimentation to show the need to and the importance of the proposed dataset. Such experimentation support the potential usegs. Even though reviewer agrees on these potentials but they are not proven and supported experimentally. Generally, there is no practical motivation to use the dataset by researchers in future.

**Additional Feedback:**

The dataset is interesting. The main comment is that several demos need to be experimented with, in order to motivate the usage of the data, in addition to the experimental comparison of the previous datasets.

**Clarity:**

The paper needs a couple of improvements here and there. For example, the Introduction section does make a strong and clear claim of the contribution leaving the reader guessing about what to read. Figure 4 does not convey any information leaving questions about why to fill this much space. Figure 5 is not readable. Equation 1 is not strongly connected to the text and not written in a compact way.

**Documentation:**

The dataset is documented properly and made available to the public.

**Ethics:**

The authors claim that there is no ethical concern.

**Relation To Prior Work:**

The dataset makes an advanced step beyond the existing datasets. It is well connected to the prior work.

**Summary And Contributions:**

The paper introduces a novel dataset of depth maps and stereo images called SurDis, it exemplifies the issue of surface discontinuity in the urban areas of Klang Valley, Malaysia.

The dataset contains:

1- more than 17000 depth maps generated from 200 sets of stereo image sequences,

2- annotations of surface discontinuity in the depth maps, and

3- bitmap stereo image pairs corresponding to the depth maps in (1).

---

> ### Author Response · Authors · 2022-08-29
> **Revised version is made accordingly**
>
> We thanks the reviewers for their valuable time spent, and for offering very constructive comments to help us improve the paper’s quality.
>
> In response to Weaknesses and Additional Feedback:
> We accepted the suggestion and added some contents about experiment and analysis.
>
> We actually did some intensive experiments and trained a range of models prior to the submission of the manuscript. However, due to page limit and we have much to share about the very unique data collection process, as well as to allow for more descriptions of the instrument developed for data collection, we thought of leaving the experiments out. We have seen a few papers with a focus on dataset from NeurIPS 2021 or earlier without experiments or benchmark, for instance [1]. Thus it became our point of reference that we could trade-off the experiment part.
>
> However, after receiving similar comments from several reviewers, in which they pointed out the importance of sharing the experiment, we have now selected a few experiments and added them to the revised version. We have also updated our Github site with the code for model training and data loading, configurations to reproduce the results were documented too.
>
> In response to Clarity:
> Figure 4 is the geographical location of the sampling, we have now removed it to make space for several other contents suggested by some reviewers.
>
> Figure 5 shows examples of noise from the raw data and they are basically darken background with some bright strokes. We decided to remove them too thanks to reviewer’s comment, since the text area should offer enough description and this will give us extra space for other more critical contents.
>
> Equation 1 was not strongly aligned with the text (thanks for pointing out) and we have amended it.
>
>
> [1] https://datasets-benchmarks-proceedings.neurips.cc/paper/2021/hash/013d407166ec4fa56eb1e1f8cbe183b9-Abstract-round1.html

---

### Meta-Review · Area_Chair_cjjm · 2022-09-10

**Recommendation:** Accept
**Confidence:** 3

**Metareview:**

All reviewers recommend that the paper be accepted. The AC sees no basis to overturn reviews, and recommends that the paper by accepted. The authors should attend to main points in the reviews when preparing the paper's final version. This includes Reviewer S4AE's suggestions about addressing diversity and balance, and Reviewer CSu3's suggestion about evaluating recent object detectors, and Reviewer Vbjb's questions about examples.

---

### Decision · Program_Chairs · 2022-09-16

Accept